# Neuroergonomics: A Perspective from Neuropsychology, with a Proposal about Workload

**DOI:** 10.3390/brainsci11050647

**Published:** 2021-05-15

**Authors:** David J. Hardy

**Affiliations:** 1Department of Psychology, Loyola Marymount University, 1 LMU Drive, Los Angeles, CA 90045, USA; david.hardy@lmu.edu; 2Department of Psychiatry and Biobehavioral Sciences, David Geffen School of Medicine, University of California, Los Angeles, CA 90045, USA

**Keywords:** neuroergonomics, neuropsychology, workload

## Abstract

In a brief overview of neuroergonomics, including some personal reminiscences of Raja Parasuraman, it is recognized that the field of human factors and ergonomics has benefitted greatly from the inclusion and integration of neuroscientific methods and theory. It is argued that such synergistic success can work in the other direction as well with the inclusion of methods and theory of human factors by a neuro field, in this case, neuropsychology. More specifically, it is proposed that neuropsychology can benefit from the inclusion of workload measures and theory. Preliminary studies on older adults, persons living with HIV, and patients with a traumatic brain injury or multiple sclerosis, are reviewed. As an adjunct measure to neuropsychological tests, the construct of workload seems perfectly suited to provide an additional vector of information on patient status, capturing some of the large individual differences evident in clinical populations and facilitating the early detection of cognitive change.

## 1. The Emergence of Neuroergonomics

Raja Parasuraman’s laboratory was surprisingly diverse in personnel and in the research topics under investigation. As a graduate student, my research focused on event-related brain potential correlates to various aspects of visual-spatial attention [1], but some of the others pursued projects involving computer programs and measures such as the Multi-Attribute Task [2], the NASA Task Load Index [3], an air traffic control simulator, and heavier equipment such as a single prop fixed-wing flight simulator. Indeed, at the time, the graduate degree you earned there was a Ph.D. in Applied-Experimental Psychology, emphasizing the applied nature of much of the research. On top of all this, Raja also spent time at the National Institutes of Health in nearby Bethesda, Maryland, collaborating on functional brain imaging projects of cognition, e.g., [4]. The lab environment was international, multicultural, interdisciplinary, and flexible. For instance, Raja had no problem at the time with me publishing a rather long paper on the cognition of older aircraft pilots [5], a topic having *nothing* directly related to my ongoing dissertation project. However, that is how Raja was, and it was an exciting place to be. With such a diverse research portfolio and strong interest in application, it is no surprise that several years later in 2007, he co-edited with Matthew Rizzo the pioneering volume *Neuroergonomics: The Brain at Work* [6]. As far as I can tell, Raja coined the term *neuroergonomics* almost ten years earlier. In a 2003 paper [7], he cites a page from his lab website dated 1998, “Neuroergonomics: The Study of Brain and Behavior at Work” (no longer active, the site was listed as http:/arts-sciences.cua.edu/csl/neuroerg.htm). Researchers rightly claim that “Raja Parasuraman’s pioneering work led to the emergence of Neuroergonomics as a new scientific field.” [8] (p. 7). The word *emergence* seems to be the proper word here, rather than *creation* or *discovery*. There was a lot of relevant research taking place at the time, within and without his own projects, but he had the creativity, experience, and vision to pull it all together, and indicate the need for a comprehensive view, a theoretical and conceptual framework to further develop and advance this growing body of work.

Neuroergonomics can be simply defined as “the study of brain and behavior at work”, with a focus on “investigations of the neural bases of such perceptual and cognitive functions as seeing, hearing, attending, remembering, deciding and planning in relation to technologies and settings in the real world.” [7] (p. 5). There are many fine papers in this special issue of *Brain Sciences*, and a comprehensive review of neuroergonomics research is not necessary here. I am not the right person to do a proper review anyway. Nonetheless, I present in Table 1 a simple analysis, listing chapter or paper titles from two volumes, the edited book by Parasuraman and Rizzo [6] and the edited book (a large collection of papers in the journal *Frontiers in Human Neuroscience*) ten years later by Gramann et al. [9]. Both of these collections serve as useful “snapshots” of the state of neuroergonomics at these two time points. In the introductory chapter [10] of the Parasuraman and Rizzo book [6], they make the claim that “a coherent body of concepts and empirical evidence that constitutes neuroergonomics theory does not exist.” (p. 6), a claim made, verbatim, four years earlier [7]. On the other hand, it is also claimed in the concluding chapter that there is “strong evidence for the growth and development of neuroergonomics since its inception” and that “Better understanding of brain function is leading to the development and refinement of theory in neuroergonomics” [11] (p. 381). One way or the other, it is clear in Table 1 that ten years later, in 2017, topics and techniques have further developed and advanced. Although the open-access platform of *Frontiers in Human Neuroscience* might be contributing to the voluminous nature of this collected volume (making a somewhat awkward-looking Table 1), the fact that such a collection could be successfully gathered speaks, in my view, to the success of neuroergonomics. Another claim in Parasuraman and Rizzo [6] is that “An imminent challenge in neuroergonomics will be to disseminate and advance new methods for measuring human performance and physiology in natural and naturalistic settings.” [11] (p. 382). It appears that researchers in neuroergonomics are vigorously engaging with this challenge. Advances are evident in the assessment of brain states and processes, especially with electroencephalography (EEG) and functional near infrared spectroscopy (fNIRS), and also with the coupling together of established methods such as pupillometry, the electrocardiogram (ECG), functional magnetic resonance imaging (fMRI), and others. Some of these advancements, as the editors of this 2017 collection note, are enabling more neuroergonomics research to move “into the wild” [8] (p. 8). Studies of brain stimulation are also interesting, such as with transcranial direct current stimulation (tDCS) in various training situations, as well as developments in brain computer interface (BDI) systems, where the brain has some control of a system, something of value to performance, and rehabilitation scenarios. Something of particular interest involves the topic of workload, especially mental workload, which seems to have waxed and waned over the years and is waxing again. This can be illustrated with subsequent editions of the substantial *Handbook of Human Factors and Ergonomics* edited by Gavriel Salvendy. In the second edition, the title of the chapter by Tsang and Wilson is “Mental Workload” [12]. In the third edition, Tsang and Vidulich changed the title to “Mental Workload and Situation Awareness” [13] where it remained as thus in the fourth edition in 2012 [14], with some discussion about whether other constructs, such as situation awareness, had supplanted or subsumed mental workload. With advancements in technology and analytic techniques, with the rise of neuroergonomics, workload and mental workload seem to be front and center once again, and this is clearly evident in the Gramann et al. collection from 2017 [9]. Workload is of central interest in the present paper.

## 2. Neuroergonomics and Neuropsychology

Neuroergonomics and neuropsychology are related in that they both, broadly construed, involve the *application* of psychological science and neuroscience to real-world situations. In neuroergonomics, the focus is on the human “operator”, the human at work. Neuropsychology focuses on the human patient, the potentially damaged or altered nervous system. As Parasuraman and Rizzo state, “Both neuropsychology and neuroergonomics rely on principles of reliability (how repeatable a behavioral measure is) and validity (what a measure really shows about human brain and behavior)” [10] (p. 8).

With regard to psychometric issues, including the standardization of tests and assessment measures, the field of neuropsychology is fairly developed, at least when more clinical measures are being used which is often the case. In general, there is agreement on the cognitive ability being assessed by a particular neuropsychological test, at least at the level of categorical domain. For instance, the Wisconsin Card Sorting Test [15] and Tower of Hanoi Test [16] assess various aspects of executive functioning and problem solving. The Hopkins Verbal Learning Test [17], Rey Auditory Verbal Learning Test [18], and California Verbal Learning Test [19] assess verbal learning and memory. A similar situation exists for many if not most neuropsychological tests and their respective domains [20]. Likewise, test performance is often scored using standardized test norms, e.g., [21]. Such norms place the group or patient test score on a scale of normalcy, indicating degree of impairment (or no impairment), in addition to accounting for factors such as age, level of education, sex, language, and sometimes even ethnicity or race. Such psychometric consistency does not really apply to most neuropsychological research using more experimentally derived tasks of cognition. For example, in a study of the human immunodeficiency virus (HIV) and early visual processing capacity [22], HIV-positive adults performed a computerized task based on an earlier basic experimental task of visual processing by Estes and Taylor [23,24]. Like most experimentalists, the psychologists who created the original task were not interested in the time-consuming development of test norms—their task was not developed to be a repeatable clinical test, including for the assessment of individuals—but rather, the refining of this procedure to better assess what they were interested in (fundamental parameters of early visual processing). In these cases, therefore, and somewhat akin to neuroergonomic research, new tasks are developed, or task paradigms are borrowed from cognitive psychology, cognitive neuroscience, and comparable disciplines, usually without the psychometric benefits of standardized test norms.

Although technological and methodological aspects of assessment have seen solid advancements in neuroergonomics, as briefly noted in the previous subsection, the establishment of guidelines, standards, and normative procedures in this field is arguably underdeveloped. There are some straightforward reasons for this. As with the more experimental research in neuropsychology, there is a natural tension between the more static clinical interest for reliable and standardized testing techniques versus the more dynamic and expansive research interests to create new tests, tasks, and procedures. Although neuroergonomics, like neuropsychology, is an applied field to varying degree, there is less emphasis on a single individual’s test results (that require comparative test norms) than in neuropsychology where there is a strong clinical aspect to the discipline. Thus, while neuropsychology can sometimes be seen as a bit static in terms of development relative to relevant informing disciplines in neuroscience, technology, and psychometrics [25], human factors and neuroergonomics might have a complimentary but perhaps less pressing issue, that of the establishment of standardized procedures and normative assessment data.

## 3. A Two-Way Street: Including Workload in Neuropsychology

The field of human factors and ergonomics has clearly benefitted from the committed inclusion of neuro-based methods and techniques and it continues to develop and advance in a variety of interesting ways. Hence, this current special journal issue. However, such a synergistic relationship can also work in the reverse direction, where the methods and measures of human factors and neuroergonomics can benefit other disciplines. For instance, in 2018, Matthew Wright and I proposed the inclusion of workload as an adjunct measure in the field of neuropsychology, to be used in tandem with tests of cognitive and motor functioning [26]. Surprisingly, at the time of our proposal, as far as we could tell, the concept of workload had never been utilized in neuropsychology, either for clinical purposes or even in research. Conducting a search in PsychINFO using the keyword *mental workload* and the more inclusive term *workload* in several of the top neuropsychology journals (*Neuropsychology*, *Neuropsychologia*, *Journal of Clinical and Experimental Neuropsychology*, and *Journal of the International Neuropsychological Society*), no citations were found. I did a similar search in 2009 with the same result (obviously) and stated this in a neuropsychology presentation [27]. I mention this because neuropsychology’s blindspot with workload, while curious, is apparently easy to overlook. As one reviewer mentioned in his or her anonymous review of our 2018 paper, “The lack of previous work on this topic somehow surprised me”. I have found more recently a thorough review published in 2017 of physiological measures of workload with a specific focus on older adults, including those with mild cognitive impairment and Alzheimer’s disease [28]. This was not published in a neuropsychology journal and seems more oriented toward basic neurocognitive research. No matter, the idea in both of these papers [26,28] is that the assessment of workload, especially cognitive or mental workload, could be useful in the characterization of the patient or clinical group such as when “two individuals or groups perform equivalently on a test, but one of them is reporting much greater mental workload (or effort, frustration, etc.)” [26] (p. 1022), and in the detection of “early cognitive decline even before their manifestation in everyday behavior” [28] (p. 516).

Like many concepts in psychology, there is no single agreed-upon definition of workload, but a useful and popular one is that it is “a hypothetical construct that represents the cost incurred by the human operator to achieve a particular level of performance” [29] (p. 140). The term *cost* refers to the idea that processing resources are limited and that successful performance on a task requires some of these resources. The more specific concept of cognitive resources is especially relevant here and relates to mental workload. A major concern in human factors and neuroergonomics is the differentiation or relationship between the performance or behavioral state of the human at work (the aircraft pilot, machine operator, etc.) and this operator’s internal state with regards to mental demand, effort, degree of frustration, and so on. The point here then is that this assessment of the relationship between external and internal states of the individual could also benefit assessments in neuropsychology. Meaning that, it might be helpful in neuropsychology to determine neurocognitive status not only based on outward test performance or brain measures, but also on workload. An illustration is provided with a popular subjective measure of workload, the NASA Task Load Index (NASA-TLX).

Originally developed by Hart and Staveland [3,29], the NASA-TLX is a self-report measure of workload that includes six subscales: Mental Demand, Physical Demand, Temporal Demand, Effort, Frustration, and Performance. For each subscale, scores range from 0 (very low) to 100 (very high). Although a weighting procedure exists for the scoring of each subscale, raw scores are often used because of ease of assessment and the finding that such an approach can be comparable to the weighted scores [29]. Overall workload can be calculated as the average of the six subscales. The NASA-TLX is typically used in conjunction with simulators or tasks that emulate real-world scenarios such as flying an aircraft, driving a car, operating a tank, and the like, to assess dimensions of workload in these situations. In contrast, we wished to validate the NASA-TLX with a typical neuropsychological test [26]. We used a test of problem solving and executive function, a computerized version of the Tower of Hanoi (TOH). The problem space in the TOH is manipulated by the number of disks involved. More disks mean more moves and more possible move combinations. In a large sample of college students (*n* = 174), we showed that workload (using raw scores) increased in a stepwise fashion from the easiest (three disks), moderate (four disks), to the most challenging TOH condition (five disks). Furthermore, subscales showing the greatest change were Mental Demand and Effort, with Physical Demand showing the smallest change. Effect sizes were quite large. Importantly, looking at individual differences, worse performance on the TOH (more total moves required to complete the TOH puzzle and greater completion times) was significantly associated with greater self-reported workload *within* each of the TOH conditions. Therefore, the NASA-TLX proved to be quite sensitive not only to task demands but also to individual differences in subjective workload. In a more recent paper, Devos and colleagues [30] examined a group of healthy older adults, where overall NASA-TLX scores were shown to be sensitive to the differential demands of an n-back task, a computerized test of working memory that while not used clinically, it is used in neuropsychological research. As working memory demands increased in the 0-back, 1-back, and 2-back conditions, overall workload increased systematically as well from 19.51 (*SD* = 15.95), 28.24 (*SD* = 17.80), to 50.92 (*SD* = 19.41). NASA-TLX overall workload also showed moderate but significant correlations with the P300 or P3 event-related brain potential, an electrophysiological marker related to workload and cognitive status. Related to the issue of aging, in an upcoming presentation from my laboratory [31], we found NASA-TLX workload ratings in an n-back task (with all assessments done online due to COVID-19 restrictions) comparable to that of Devos et al. [30], but we included a younger adult comparison group and found greater Mental Demand and Effort subscale ratings in the older adult group (*n* = 42) compared to the younger group (*n* = 32) but only in the most difficult n-back condition (3-back). This older group (between 54 and 76 years old) was not really a clinical group, but the NASA-TLX group difference evident in the challenging 3-back condition (with no group difference in response accuracy, the older group did show slower reaction times) suggests, possibly, an early sign of mild cognitive decline. Workload has been examined in a few clinical populations. In an early small-scale study funded by the National Institute on Aging [27], a small group of adults with HIV completed the NASA-TLX in relation to performance on the Multi-Attribute Task (MAT) [2]. The MAT, commonly used in human factors research but not in neuropsychology, can be viewed as a low-fidelity flight simulation task involving a tracking task, fuel management task, and systems monitoring task. The MAT is an ideal instrument to measure multi-tasking ability, something shown to be impaired in some persons living with HIV [32,33,34]. HIV-positive participants (*N* = 32) performed a tracking task alone, in a dual-task condition, and in a tri-task condition (in combination with the fuel management and system monitoring tasks). Workload was assessed in the middle of each blocked tracking condition. Focusing on the Mental Demand subscale, we found an expected stepwise increase in mean Mental Demand scores in the single (*M* = 56.25, *SD* = 24.36), dual (*M* = 64.44, *SD* = 22.52), and tri-task (*M* = 71.63, *SD* = 22.14) tracking conditions (*p* < 0.001). To assess mental workload and its relationship with task performance, we divided participants into low and high mental workload groups based on a median split in Mental Demand scores in the single tracking condition. The difference in Mental Demand between low (*M* = 37.82, *SD* = 17.23) and high (*M* = 77.13, *SD* = 9.75) groups was, as expected, fairly large (*p* < 0.001). More interesting was the finding that tracking performance, measured as mean root mean square error (RMSE) so larger scores indicate worse performance, in the single task condition was *not* different between the two groups (see Figure 1). This is relevant to the present proposal, that workload can be a useful adjunct measure in neuropsychology, because in this particular case, it is illustrated that subjective workload can vary independently of task performance. The NASA-TLX captured a source of variability, one that is relevant to the assessment of cognitive status, that would have been otherwise unknown. A group difference in tracking performance did become more apparent with an increase in task challenge due to divided attention and/or multi-tasking, although this was only a trend (*p* = 0.11), presumably due to the small sample sizes and low statistical power. A proposed conclusion therefore is that although the two groups were comparable in task performance, especially in the single-task tracking condition, the cognitive status of the two groups was actually not comparable due to the increased subjective workload in the high Mental Demand group. That this group showed worse performance (at a trend level) when task demands increased in the dual and tri-task conditions offers tentative support to this interpretation. A similar result was demonstrated in a study of patients with traumatic brain injury (TBI) [35]. Here, the TBI (*n* = 20) and healthy control (*n* = 32) groups performed a traditional neuropsychological battery of paper-and-pencil tests across a variety of cognitive domains. After each test, participants completed the NASA-TLX. One result of interest was that although there was no significant group difference in the Visuospatial Ability domain (*p* = 0.617), the TBI group did tend to report higher levels of workload (but not at an alpha level of 0.05) compared to the control group (see Figure 2). And as expected, Physical Demand showed the lowest overall levels of reported workload as well as the smallest group difference, which makes sense because the test that this domain was based on (the Hooper Visual Organization Test) is not physically demanding. As with the previous HIV study described [20], here is an illustration where groups do not differ on a behavioral cognitive test but where subject workload differences suggest perhaps a group difference in cognitive status, where the TBI individuals needed to recruit greater cognitive resources, a compensatory strategy, to maintain comparable performance. In contrast to these findings was a report on a small group (*n* = 10) of cognitively impaired adults with multiple sclerosis [36]. In this study, although this group performed worse on a variety of neuropsychological tests compared to a cognitively unimpaired multiple sclerosis group (*n* = 12) and healthy controls (*n* = 22), there were no group differences in subjective workload (NASA-TLX) nor in pupillary size (a psychophysiological measure of cognitive workload). However, interpreting the findings in this study is difficult. It is possible that the cognitively impaired multiple sclerosis group did not need to recruit greater effort or resources, or perhaps were deficient in such compensatory processes. However, when observing mean workload scores (which the authors generously provided for every single measure in their Table 1), the control group’s NASA-TLX scores are usually much closer to the impaired group’s ratings while the cognitively unimpaired multiple sclerosis group had the lowest workload ratings. Although there is nothing obviously unique about the healthy control group, perhaps this issue needs a closer examination? Furthermore, this is a recent report, and hopefully participants in all three groups will continue to be recruited for this study.

It is clear that the application of workload in neuropsychology, especially with subjective measures such as the NASA-TLX, is in the early stages of development. Hence, the need to repeat this proposal. The Hardy and Wright paper [26], although published in a solid neuropsychology journal in 2018, as of this writing has been cited only six times according to Google Scholar, none of them being neuropsychological studies or in a neuropsychology journal. So, there is work to be done. Although basic psychophysiological and brain imaging research that can be related to workload is more developed, especially within the population of older adults [28], much of the more neuropsychologically or clinically oriented data presented here are admittedly preliminary in nature. Further, there are other measures of workload to consider of course in various modalities (self-report, behavioral, physiological, etc.). That said, it is argued that results so far are positive enough to warrant further efforts in this pursuit. The inclusion of workload measures in neuropsychological assessments also makes sense from a larger theoretical or conceptual perspective. Thirty years ago, a National Institute of Mental Health (NIMH) workshop urged for the use of “new experimental paradigms borrowed from cognitive psychology” in neuropsychological assessments [37] (p. 964). The proposal here can be viewed as a natural or evolving appendage to this recommendation, that neuropsychology could also benefit from the use of paradigms from human factors and neuroergonomics [26]. Furthermore, this older NIMH recommendation has been extensively modified and broadened in scope with the more recent NIMH initiative referred to as RDoC (Research Domain Criteria) [38]. In the RDoC initiative, the emphasis is on the investigation of basic biological and cognitive processes (with a focus on the interacting domains of negative valence systems, positive valence systems, cognitive systems, systems for social processes, and arousal/regulatory systems) and their role, how they are modified, in mental health and pathology. Part of the rationale for RDoC is that such an approach will better capture and explain the large variability always evident in individuals within traditional diagnostic categories. It is argued that the workload construct can be nicely situated within this framework, to help in the delineation of the cognitive and neural status of patients with varying degrees of neuropathology and ever present large individual differences.

## 4. Conclusions

Research and disciplines have their boundaries, and often for good reasons. But sometimes it is helpful to soften and cross such boundaries. Neuroergonomics is a case in point. It is clear that the field of human factors/ergonomics has benefitted with the inclusion and integration of neuroscientific methods and theory. However, this exchange can work in both directions. Although not the main focus of this Special Issue of *Brain Sciences*, what I have tried to do here is briefly show how methods and theory of human factors and neuroergonomics can also potentially enhance a neuro field, in this case, neuropsychology. Specifically, the inclusion of workload as an adjunct measure to the tests of neuropsychology can provide an additional and complimentary vector of information on the neurocognitive status of the patient. This also applies to more experimental research in neuropsychology. Workload can provide a novel mechanism in the understanding of neurocognitive sequalae and also help capture the large individual differences evident in clinical populations. Although importing measures, approaches, and constructs from one field (human factors and neuroergonomics) into another field (neuropsychology) can result in a more complex scenario—now in neuropsychology, we relate cognition and behavior not only to brain state and function but also include the mediating construct of workload to this montage—the evolving field of neuroergonomics shows how there can be great potential payoff as well. In a sense, this forces you to be an interdisciplinarian. It also fits nicely within the current NIMH RDoC initiative [38] which is trying to advance and develop approaches to mental health research broadly construed.

As I began this paper, I would like to end it on a personal note. Thinking about interdisciplinarianism, Raja, a native of India, spent his college years in England. Both of us avid racquetball players, I was the younger man with more power in my shots, but he was tricky with these unusual finesse shots he learned while playing squash in the UK. And he would win! An interdisciplinarian in more ways than one! He was a brilliant researcher and mentor, and I would like to think he would approve of the workload proposal in this paper. In addition to a brilliant mind, Raja was a generous host, he loved food and fun. He was always up for the pub, pints, and the occasional cigarette, music, the pool table. I recall a dinner at his house when he lived in the bohemian Adams Morgan neighborhood of Washington, D.C., where I sat speechless for an hour, not because sitting at the table was Michael Posner, the pioneering psychologist and cognitive neuroscientist, but because I had eaten an innocent-looking Indian pickled pepper, with mouth aflame and palpitating heart just trying to maintain composure. Sadly, as many know, Raja died unexpectedly in 2015. The edited collection in 2017 by Gramann et al. [9] was thoughtfully dedicated to him. I never did pay him tribute publicly, so I am glad to do so now as part of this review.

## Figures and Tables

**Figure 1 brainsci-11-00647-f001:**
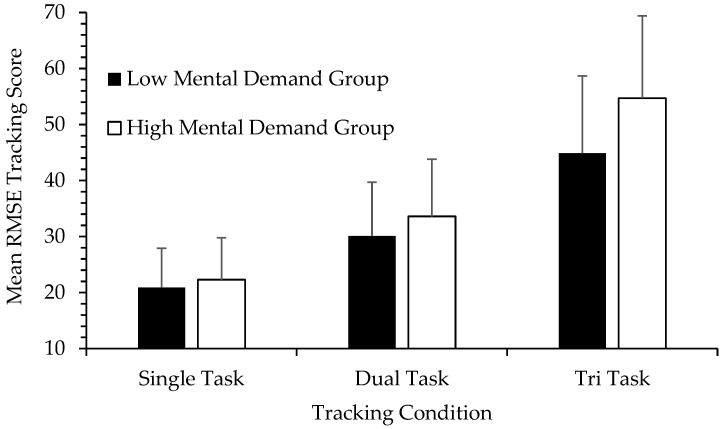
Tracking performance (mean root mean square error (RMSE) and 95% confidence internals) of HIV-positive adults on the Multi-Attribute Task [20]. RMSE, a measure of error variability, indicates worse performance with larger scores.

**Figure 2 brainsci-11-00647-f002:**
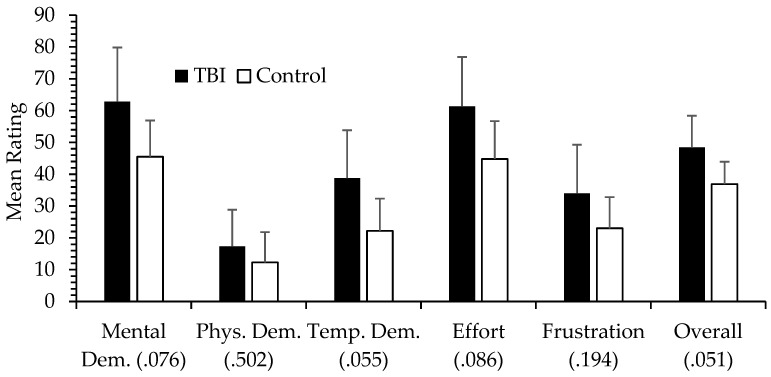
Mean NASA-TLX workload scores (and 95% confidence internals) for the TBI and control groups in the visuospatial neuropsychological domain [28]. Parentheses include ANOVA *p* values.

**Table 1 brainsci-11-00647-t001:** Comparison of topics in *Neuroergonomics: The Brain at Work* edited by Parasuraman and Rizzo [6] and *Trends in Neuroergonomics: A Comprehensive Overview* edited by Gramann et al. [8].

Parasuraman and Rizzo (2007)	Gramann et al. (2017) ^1^
Introduction to Neuroergonomics	Introduction
Electroencephalography (EEG) in Neuroergonomics Event-Related Potentials (ERPs) in Neuroergonomics Functional Magnetic Resonance Imaging (fMRI): Advanced Methods and Applications to Driving Optical Imaging of Brain Function Transcranial Doppler Sonography Eye Movements as a Window on Perception and Cognition The Brain in the Wild: Tracking Human Behavior in Natural and Naturalistic Settings Spatial Navigation Cerebral Hemodynamics and Vigilance Executive Functions The Neurology of Emotions and Feelings, and Their Role in Behavioral Decision Stress and Neuroergonomics Sleep and Circadian Control of Neurobehavioral Functions Physical Neuroergonomics Adaptive Automation Virtual Reality and Neuroergonomics The Role of Emotion-Inspired Abilities in Relational Robots Neural Engineering EEG-Based Brain-Computer Interface Artificial Vision Neurorehabilitation Robotics and Neuroprosthetics Medical Safety and Neuroergonomics Future Prospects for Neuroergonomics	Mobile Brain/Body Imaging (MoBI) of Physical Interaction with Dynamically Moving Objects Age-Sensitive Effects of Enduring Work with Alternating Cognitive and Physical Load. A Study Applying Mobile EEG in a Real-Life Working Scenario Benefits of Instructed Responding in Manual Assembly Tasks: An ERP Approach Pre-Trial EEG-Based Single-Trial Motor Performance Prediction to Enhance Neuroergonomics for a Hand Force Task Evaluation of a Dry EEG System for Application of Passive Brain–Computer Interfaces in Autonomous Driving An Intelligent Man-Machine Interface—Multi-Robot Control Adapted for Task Engagement Based on Single-Trial Detectability of P300 Perception and Cognition of Cues Used in Synchronous Brain–Computer Interfaces Modify Electroencephalographic Patterns of Control Tasks The Brain is Faster than the Hand in Split-Second Intentions to Respond to an Impending Hazard: A Simulation of Neuroadaptive Automation to Speed Recovery to Perturbation in Flight Attitude Efficient Workload Classification based on Ignored Auditory Probes: A Proof of Concept Gaussian Process Regression for Predictive But Interpretable Machine Learning Models: An Example of Predicting Mental Workload Across Tasks Evaluation of an Adaptive Game that Uses EEG Measures Validated during the Design Process as Inputs to a Biocybernetic Loop Neural Mechanisms of Inhibitory Response in a Battlefield Scenario: A Simultaneous fMRI-EEG Study Exploring Neuro-Physiological Correlates of Drivers’ Mental Fatigue Caused by Sleep Deprivation Using Simultaneous EEG, ECG, and fNIRS Data Steering Demands Diminish the Early-P3, Late-P3, and RON Components of the Event-Related Potential of Task-Irrelevant Environmental Sounds Toward a Wireless Open Source Instrument: Functional Near-infrared Spectroscopy in Mobile Neuroergonomics and BCI Applications Why a Comprehensive Understanding of Mental Workload through the Measurement of Neurovascular Coupling Is a Key Issue for Neuroergonomics? Acute Supramaximal Exercise Increases the Brain Oxygenation in Relation to Cognitive Workload Prefrontal Cortex Activation Upon a Demanding Virtual Hand-Controlled Task: A New Frontier for Neuroergonomics Into the Wild: Neuroergonomic Differentiation of Hand-Held and Augmented Reality Wearable Displays during Outdoor Navigation with Functional Near Infrared Spectroscopy Processing Functional Near Infrared Spectroscopy Signal with a Kalman Filter to Assess Working Memory during Simulated Flight Commentary: Cumulative effects of anodal and priming cathodal tDCS on pegboard test performance and motor cortical excitability Simultaneous tDCS-fMRI Identifies Resting State Networks Correlated with Visual Search Enhancement Transcranial Direct Current Stimulation Modulates Neuronal Activity and Learning in Pilot Training Does a Combination of Virtual Reality, Neuromodulation, and Neuroimaging Provide a Comprehensive Platform for Neurorehabilitation? A Narrative Review of the Literature Corrigendum: Does a Combination of Virtual Reality, Neuromodulation, and Neuroimaging Provide a Comprehensive Platform for Neurorehabilitation? A Narrative Review of the Literature High Working Memory Load Impairs Language Processing During a Simulated Piloting Task: An ERP and Pupillometry Study The Impact of Expert Visual Guidance on Trainee Visual Search Strategy, Visual Attention, and Motor Skills The Role of Cognitive and Perceptual Loads in Inattentional Deafness From Trust in Automation to Decision Neuroscience: Applying Cognitive Neuroscience Methods to Understand and Improve Interaction Decisions Involved in Human Automation Interaction

^1^ Individual citations in this volume have a 2016 publication date for some reason.

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
