# Peer review of "Neuroergonomics: A Perspective from Neuropsychology, with a Proposal about Workload"

_brainsci, 2021, doi:10.3390/brainsci11050647_

Round 1
Reviewer 1 Report
This brief review gives a nice introduction and background to Raja Parasuraman’s et al. work with the concept workload. The author makes an interesting and valid point about measuring workload in neuropsychological testing. Empirical data are presented as illustrative examples and in support of the main argument.
Author Response
Thank you for the positive comments. Changes in the revised manuscript are in track changes.
Reviewer 2 Report
The paper chronicles the emergence and development of the field of Neuroergonomics and highlights that the field, in part, draws heavily on physiological and Neurophysiological measurement techniques. It relates Neuroergomics with Neuropsychology focussing on the role of psychometric testing and the use of tasks to assess cognition. The paper goes on to describe the thesis that workload as measured, for example, by the NASA Task Load Index (NASA TLX), a construct associated with Neuroergonomics, can make a useful contribution to assessing cognitive decline in Neurophysiology. I believe that a major purpose of the paper is to highlight that thesis which may have previously received less attention but is put forward in the self-cited papers (19, 20, and 28) each of which describe an individual lab study.
The strengths of the paper include its relaxed narrative style and the informative section 1 which acts as an interesting introduction. Section 2 does a good job of framing the relationship between Neuroergonomics and Neurophysiology such that the scene is set for the description of the main thesis that the paper is highlighting. Another strength is that, this thesis that workload measurement might be used to predict cognitive decline is, I believe, a valid one and is worthy of further investigation.
The main weakness of the paper is that the main thesis that the paper is promoting has already been proposed in the three self-cited lab study papers and is therefore not, in itself, completely new. Sadly, I was unable to find citations 20 and 28 on-line but 19 was readily available.
However, it is a strength that this review paper brings together the three self-cited lab studies, presenting the results from the two, difficult to obtain, poster presentations (20 and 28) and uses them together to further argue for the thesis.
An easily rectifiable weakness is the lack of clear exemplification of the statement that “One way or another it is clear in Table 1 that ten years later… techniques have .. advanced” (line65 to 67). I felt there that perhaps picking out one or two examples from the long table and using them to make the point would have helped. Another possible (but relatively minor) weakness, for me, was the description of a keyword search in PsychINFO for the terms workload and ‘mental workload’ not uncovering any citations (lines 153 to 158). I was wondering if the term ‘cognitive load’ been used perhaps there might have been results?
Suggestions for improvement:
I think it would be useful to include clarification that the version of the NASA TLX that was used in the cited papers was the raw unweighted version (citing reference 22).
Add some specific exemplification of the statement (lines 65 to 67) about Table 1 illustrating advances. I know that the totality of the table is part of the argument but it would help the reader if one or two items were picked out and used to individually illustrate it, perhaps by contrasting or disecting those entries.
Include citations for the neurophysiological tests listed in para 2 of section 2 (line 115 to 118).
Include clarification around lines 194 and 195 that the Tower of Hanoi game used was a computerised version (which is clear when one reads reference 19). This may seem pedantic but with the NASA TLX including a physical demand subscale this is relevant. Stating the game was computerised would remove ambiguity.
Consider getting ‘Workload’ and the thesis that measuring workload can help predict cognitive decline, into the title to make the title more representative of the paper as a whole. At the moment the title really only describes half of the paper.
Summary:
I found the paper an enjoyable and informative read. Although the weight of contribution is not great, I believe there is value in bringing together the earlier lab studies and therefore it is worthy of publication.
As a codicil: It was nice to see the author pay homage to the late R.Parasuraman.
Author Response
I appreciate the critical and constructive comments from Reviewer 2.
In response to the comment that the main weakness of the paper is that the main thesis has already been proposed, new text has been inserted (in the second paragraph below Figure 2, starting with the sentence "Hence the need to repeat this proposal. The Hardy and . . . ." As mentioned in this new text, nobody in neuropsychology has cited my 2018 paper. This is not mentioned in this statement, but it's also the case that my other two references are poster presentations so it's difficult to get details (including the present proposal) on this as the reviewers have found out for themselves.
In reply to the comment about the "lack of clear exemplification of the statement that "One way or the other it is clear in Table 1 . . . .", my sense is that the comments that come after Table 1 illustrate this point enough since this isn't really the main point of the paper. On the other hand, if Review 2 continues to consider this a strong point, then I'll be happy to include more detail.
With regards to searching under 'cognitive load', some hits are found but these relate to task difficulty, not workload or mental workload.
I now note the use of raw unweighted NASA-TLX scores (see the third paragraph in section 3; I don't mention lines here because these are all messed up at this point because of track changes).
References are now included for the neuropsychological tests listed in the second paragraph in section 2.
It is now mentioned that the Tower of Hanoi test was computerized (see paragraph 3 in section 3).
The paper title now includes the word "workload".
Reviewer 3 Report
The article explains how human factors and ergonomics have benefitted from methodological approaches borrowed from neuroscience. It then argues for passing back insights from the human factors domain, especially the workload construct (NASA TLX to be specific), into the field of neuropsychology.
The tone of the text is straightforward and sometimes personal. That is unusual but refreshing for a major journal publication (ref l54, l69). I personally like it; it makes the text very approachable even to novices in the literature.
Table 1 mostly underlines the point “techniques have 66 further developed and advanced” l66. Herby, it occupies plenty of space. It is interesting but probably too prominent for not contributing much new information. I suggest reducing the font size (currently, items are larger than the main text), making it visually somewhat less dominant.
Chapter 2, paragraph 2 describes the saturated landscape of psychometric tools neuropsychology. In l123 the author speaks about “psychometric consistency”, but I’m not sure what this consistency is. It probably refers to potentially confounding variables mentioned before (e.g., age, education), but the reference could be clarified. Further, why are these not applicable to neuropsychological research? It is stated that this is because of the more experimentally derived tasks, but I find this chain of thought hard to follow.
L135/136: Maybe a definition of “application” would help here. I think one could argue oppositely, that neuroergonomics is more applied since it is concerned with “real-world” artifacts. Neuropsychology often uses rather abstract and somewhat detached test environments like a choice reaction time task. The statement feels more like an opinion than an evidence-backed claim.
Chapter 3 (A Two-Way Street) is particularly great! However, in l249 the author concludes that the “cognitive status” of the two groups is not comparable. This is not surprising since their expressed mental workload distinguished the groups. Maybe I miss the vital part here, but I did not understand the finding. A group of participants is divided into the best and least-best performers, and then it is concluded that they differ in their cognitive status. I suggest elaborating on the term “cognitive status”, since it is not motivated in the text so far. Maybe ever highlight the critical finding, as of now, I find it hard to grasp. As I understand, the main point is that the groups do not show differences on a behavioral, cognitive test, but their workload level. However, this should be clarified in the text, especially at which point they do not differ. To further support the understanding of that part, I suggest extending the Figure 1 labeling; mainly, the Y-axis could be more precise. A legend would help understand the color schema of the graph further.
In the description of the second study in l258, you mention that “the TBI group did tend to report higher levels”. While you provide p-values in the description of Figure 2, I expected them in the text as well (as you did for the Visuospatial Ability domain). The word “tend” indicates non-significant findings, but I suggest calling it as such for transparency. After reading these sentences and finding the p-values in the figure description, not mentioning that none of the dimensions yielded a significant difference did not feel particularly faithful. I could not read the full text of reference 28 (Inclusion of workload in neuropsychological assessment: A preliminary illustration with TBI patients) after only a quick search to look up the number of participants. Please provide this as well since it is essential for an evaluation of statistical significance.
You state that you show how methods and theories of HF can be leveraged in neuro fields (e.g. l318). For my linking, the theory part came up a bit short. I would like to encourage you to extend more theories, background, and explanations in this regard.
The last paragraph of the discussion is indeed a personal note. I enjoyed reading it. You surely thought about if the discussion of a paper – although unusual - is the right place for it. I am not saying it is not. I am saying I had to think about it.
While I do not think it is a review (but rather a well-written story about your perspective), I found the manuscript interesting to read. Most thoughts are easy to follow, and your main message is clear. My two key points are the general comprehensibility of the experiment around Figure 1 and a minor thing about the precision of the statistical reporting around l258.
Minor details:
- Some commas seem to be missing in the following lines:
- L24, “As a graduate student, …”
- L106, “In neuroergonomics, …”
- L146, “Hence, …”
- L 171, “Like many concepts in psychology, …”
- L254, “Here, …”
- L267 probably uses a wrong parenthesis “… HIV study described [20), …”
Author Response
Thanks to Reviewer 3 for the critical and helpful comments. I realize the tone or style of the paper is a bit unusual, but that was my intention, so I'm happy to see that it worked, at least with this reviewer.
With regards to Table 1, the font size does not appear to me to be larger than the regular text font. Furthermore, I'm not sure if font size can be reduced or not. That seems to me to be a matter of choice of the publisher.
With regards to the comment about "psychometric consistency" (in the middle of paragraph 2 in section 2), I've tried to clarify this issue with new text and a new reference.
With regards to the comment about "application", I've removed the comment about neuroergonomics being perhaps less applied. This is not a central point and I'm not really interested in sizing up and comparing the applied nature of neuroergonomics versus neuropsychology.
Addressing the questions about Figure 1 and the study that it pertains to (this is in section 3), some of the text has been reworded (all changes in this revised manuscript are in track changes) and I've recreated Figure 1. It now includes a legend, and the figure caption and Y-axis has been modified. Hopefully this is more clear now.
With regards to the statistical trends in the second study (section 3, on p. 5), a statement is not inserted to explicitly clarify this "not at an alpha level of .05". Participant group sizes are also included now.
Missing commas are now supplied. I think I was trying to keep the text moving briskly, without commas, but . . . commas are now provided. Thank you for noting this.
I agree, this paper is less of a review and more of a personal narrative. It is certainly not a comprehensive review and was not intended to be so. The reviews parts and proposal could be considered, I think, brief introductions to these topics, with a dash (maybe a large dash) of personal reminiscences about RP. I'm not exactly sure how this somewhat unorthodox approach came to me, although I think reading the Gramann et al. (2017) editorial jogged my memory about my grad school mentor. I think it works in this case. I hope others agree.